# *Bifidobacterium breve* CNCM I-4035, *Lactobacillus paracasei* CNCM I-4034 and *Lactobacillus rhamnosus* CNCM I-4036 Modulate Macrophage Gene Expression and Ameliorate Damage Markers in the Liver of Zucker-Lepr*^fa/fa^* Rats

**DOI:** 10.3390/nu13010202

**Published:** 2021-01-11

**Authors:** Luis Fontana, Julio Plaza-Díaz, Paula Robles-Bolívar, Héctor Valente-Godínez, María José Sáez-Lara, Francisco Abadía-Molina, Carolina Gómez-Llorente, Ángel Gil, Ana I. Álvarez-Mercado

**Affiliations:** 1Department of Biochemistry and Molecular Biology II, School of Pharmacy, Campus de Cartuja s/n, 18071 Granada, Spain; jrplaza@ugr.es (J.P.-D.); gomezll@ugr.es (C.G.-L.); agil@ugr.es (Á.G.); 2Institute of Nutrition and Food Technology “José Mataix”, Biomedical Research Center, Parque Tecnológico Ciencias de la Salud, Avda. del Conocimiento s/n, Armilla, 18100 Granada, Spain; mjsaez@ugr.es; 3Instituto de Investigación Biosanitaria ibs. GRANADA, Complejo Hospitalario Universitario de Granada, 18071 Granada, Spain; fmolina@ugr.es; 4Children’s Hospital of Eastern Ontario Research Institute, Ottawa, ON K1H 8L1, Canada; 5Department of Cell Biology, School of Sciences, Campus de Fuente Nueva, 18071 Granada, Spain; paurrobo@gmail.com; 6Division of Health Sciences, Campus León, Department Medicine and Nutrition, University of Guanajuato, 36000 Guanajuato, Mexico; hector_valente@live.com.mx; 7Department of Biochemistry and Molecular Biology I, School of Sciences, Campus de Fuente Nueva, 18071 Granada, Spain; 8CIBEROBN (Physiopathology of Obesity and Nutrition), Instituto de Salud Carlos III (ISCIII), 28029 Madrid, Spain

**Keywords:** NAFLD, liver damage, probiotics, inflammation, polarization, macrophages

## Abstract

Non-alcoholic fatty liver disease (NAFLD) has reached pandemic proportions worldwide. We have previously reported that the probiotic strains *Bifidobacterium breve* CNCM I-4035, *Lactobacillus paracasei* CNCM I-4034 and *Lactobacillus rhamnosus* CNCM I-4036 exert anti-inflammatory effects in the intestine of Zucker-Lepr*^fa/fa^* rats. In this work, we focused on their hepatic effects. M1 macrophages are related to inflammation and NAFLD pathogenesis, whereas M2 macrophages release anti-inflammatory mediators. We evaluated the effects of these 3 strains on macrophage polarization, inflammation and liver damage of Zucker-Lepr*^fa/fa^* rats. The animals received either a placebo or 10^10^ CFU of probiotics orally for 30 days. Nos2 and Cd86 mRNA levels were determined as markers of M1 macrophages, and Cd163 and Arg1 as M2 markers, respectively, by qRT-PCR. Liver damage was determined by lipid peroxidation, leukocyte infiltration and myeloperoxidase activity. We evaluated a panoply of circulating chemokines, the hepatic ratio P-Akt/Akt, NF-kB and P-NF-kB protein levels. All 3 probiotic strains modulated macrophage polarization in liver and circulating levels of inflammation-related mediators. *L. paracasei* CNCM I-4034 increased the ratio P-Akt/Akt and NF-kB protein levels. *B. breve* CNCM I-4035, *L. paracasei* CNCM I-4034 and *L. rhamnosus* CNCM I-4036 decreased both pro-inflammatory macrophage gene expression and leukocyte infiltration in the liver.

## 1. Introduction

Non-alcoholic fatty liver disease (NAFLD) has reached pandemic proportions worldwide [1]. This disease is characterized by fat accumulation in the form of micro and macrovacuoles of lipids in hepatocytes [2] and may progress from simple steatosis, non-alcoholic steatohepatitis to fibrosis, advanced cirrhosis and hepatocarcinoma [3]. However, its pathophysiology has not yet been fully defined.

The liver is an immunologically complex organ due to its host non-lymphoid cells including stellate and dendritic cells, and lymphoid cells, macrophage Kupffer cells. Many of these cells are components of the classic innate immune system, enabling the liver to play a major role in the response to pathogens [4]. 

Macrophages play central roles in host defense, immune regulation, tissue repair, and regeneration. They are generally delineated into 2 categories: M1 and M2 macrophages. Macrophages adapt their phenotypes in response to various microenvironmental signals, and exhibit different characteristic markers, gene expression profiles and functions [5]. In response to stimuli or polarization signals, M1 macrophages show pro-inflammatory properties and secrete pro-inflammatory mediators which induce inflammation and damage [6,7], as well as antimicrobial and antitumor resistance. In contrast, M2 macrophages are activated by IL-4/IL-13, and release anti-inflammatory or pro-resolving mediators which mediate wound repair, tissue remodelling [6] and restore functions. In addition, M2 macrophages promote wound healing, angiogenesis, and resistance to parasites [6,7]. This dichotomy is intimately associated with the great plasticity and remarkable heterogeneity of macrophages [8].

Pro- and anti-inflammatory macrophages are characterized by specific pathways that regulate the metabolism of lipids and amino acids and affect their responses [9]. Secretion of appropriate cytokine and chemokines from macrophages may lead to the modification of the microenvironment for bridging innate and adaptive immune responses [8]. In the liver, inflammation is regulated by the balance of M1 and M2 macrophages. Thus, the exacerbated release of M1-derived cytokines, such as TNF-α, and chemokines including monocyte chemoattractant protein (MCP)-1 and RANTES, stimulates fibrogenesis, thus contributing to the pathogenesis of liver disease [10]. Accordingly, since inflammatory processes are crucial in the potential progression of NAFLD, chemokines might also play a pivotal role in NAFLD pathophysiology. In fact, chemokines and their receptors have been implicated in multiple inflammatory diseases, such as atherosclerosis, multiple sclerosis, psoriasis, and insulin resistance. The expression of several chemokines and chemokine receptors is shown to be upregulated in the livers of obese patients with severe steatosis and non-alcoholic steatohepatitis (NASH) [11].

The intestinal microbiota controls and maintains the function of both the liver and the intestine [12]. Alterations in the intestinal microbiota (referred to as dysbiosis) [13] also directly influence the liver and may trigger pathologies in both organs [14]. The close relationship between these two organs (the so-called gut-liver axis) is achieved by the connection in a bidirectional way through the biliary tract, the portal vein and the systemic circulation [12]. A balanced gut microbiota is crucial for human health because of its key role in regulating the development and function of different cells of the immune system such as T lymphocytes and different pro-inflammatory factors [14]. The intestinal microbiota has also an impact on lipogenesis and thereby on steatosis [15]. Moreover, cumulative data demonstrate that dysbiosis is greatly associated with NAFLD, although the underlying mechanisms remain mostly unknown [16]. In this regard, a key role in the pathogenesis of NAFLD is played by gut-derived endotoxin, a component of the Gram-negative bacteria wall, which reaches the liver when the colonic mucosa is disturbed. Also, lipopolysaccharide (LPS) and other bacterial products such as pathogen-associated molecular patterns (PAMPs) can be released into the blood leading to the activation of the innate immune response [4].

Probiotics are live microorganisms that confer a health benefit to the host when administered in adequate amounts [17]. Probiotics are consumed to enrich and strengthen the microbiota, thereby preventing dysbiosis [18]. These microorganisms exert both immunomodulating and antitumor effects [19]. Probiotics may affect the intestinal microbiota and the mucosal immune system [20]. The use of probiotics for the modulation of the intestinal microbiota as a strategy for the treatment of NAFLD has been addressed by several authors with promising results, Bifidobacterium and Lactobacillus genera being the most commonly used in functional foods and dietary supplements [21].

Our group has worked with 3 probiotic strains (*Bifidobacterium breve* CNCM I-4035, *Lactobacillus paracasei* CNCM I-4034 and *Lactobacillus rhamnosus* CNCM I-4036) isolated from the feces of newborn infants exclusively breastfed [22]. We have reported that these probiotics are safe and showed, among others, their effect on the expression of genes associated with inflammation (*Adamdec1*, *Ednrb* and *Ptgs1*/*Cox1*) [23]. The intake of these probiotic strains also reduced hepatic steatosis, decreased serum LPS, and presented anti-inflammatory effects in obese Zucker rats [24]. Considering the aforementioned results, in the present study we aimed to elucidate the relationship between the intake of *B. breve* CNCM I-4035, *L. paracasei* CNCM I-4034 and *L. rhamnosus* CNCM I-4036 on macrophage modulation in the NAFLD model of Zucker-Lepr^fa/fa^ rats.

## 2. Materials and Methods

### 2.1. Probiotic Bacteria

The probiotic strains *Lactobacillus paracasei* CNCM I-4034, *Bifidobacterium breve* CNCM I-4035, and *Lactobacillus rhamnosus* CNCM I-4036 have been characterized and are described elsewhere [22].

### 2.2. Ethical Approval

The procedures observed the guidelines for animal research of the University of Granada (Spain) and was approved by the Ethics Committee on Animal Research of the University of Granada (Permit Number CEEA: 2011–377).

### 2.3. Experimental Animals

This study was performed using Zucker-Lepr*^fa/fa^* male rats weighing 168–180 g purchased from Harlan Laboratories (Charles River, Barcelona, Spain). All the animals were kept in metabolic cages with a 12-h light-dark cycle with free access to water and food for 5 days to adaption.

### 2.4. Experimental Design

After 5 days of adaptation, the rats were randomly divided into five groups of 8 animals each:Control group. Zucker-Lepr*^fa^^/fa^* rats that were euthanized as a reference or baseline after 5 days of adaptation (time 0).Placebo group. Zucker-Lepr*^fa^^/fa^* rats that received placebo by oral gavage for 30 days. The placebo contained 67% cow’s milk powder, 32.5% sucrose, and 0.56% vitamin C.*B. breve* group. Zucker-Lepr*^fa^^/fa^* rats that received 10^10^ colony-forming units (CFUs) of *B. breve* CNCM I-4035 by oral gavage for 30 days.*L. paracasei* group. Zucker-Lepr*^fa^^/fa^* rats that received 10^10^ CFUs of *L. paracasei* CNCM I-4034 by oral gavage for 30 days.*L. rhamnosus* group. Zucker-Lepr*^fa^^/fa^* rats that received 10^10^ CFUs of *L. rhamnosus* CNCM I-4036 by oral gavage for 30 days.

### 2.5. Sample Collection

After 5 or 30 days the animals were anesthetized and sedated with ketamine and xylazine. Blood was drawn from the aorta and centrifuged for 10 min at 1000× *g* and 4 °C to separate the serum from cells. Samples of liver and intestinal mucosa were also taken and immediately frozen in liquid nitrogen for quantitative real-time polymerase chain reaction (qRT-PCR) and Western blotting. Liver tissue samples for histology and hematoxylin-eosin staining were fixed with 4% paraformaldehyde-phosphate buffered saline for 4 h at room temperature and embedded in paraffin.

### 2.6. Histology

Paraffin-embedded liver samples were sliced onto 5 µm-thick sections and stained with hematoxylin and eosin for histological examination. Four animals per group and 5 sections per animal were microscopically analyzed to determine leukocyte infiltrate with ImageJ software using gray-scale. 

### 2.7. RNA Extraction and Real-Time Quantitative PCR (qRT-PCR)

Liver tissue RNAs were isolated using RNeasy Mini Kit (Qiagen, Barcelona, Spain). Complementary DNA (cDNA) was synthesized using the iScript advanced cDNA Synthesis Kit (Bio-Rad Laboratories, CA, USA). cDNA was amplified with SYBR Green PCR Master Mix (Applied Biosystems, Scotland, UK) and an ABI Prism 7900 instrument (Applied Biosystems, Foster City, CA, USA). The specific primers used were: Cd86, assay ID: qRnoCID0008321; Nos2, assay ID: qRnoCID0017722; Cd163, assay ID: qRnoCID0008321; Arg1, assay ID: qRnoCID0006520 (Bio-Rad, Coralville, IA). q-PCR data were normalized to the hypoxanthine phosphoribosyltransferase 1 (Hprt1) (assay ID: qRnoCED0057020, Bio-Rad, Coralville, IA). The PCR conditions were 1 cycle of 95 °C for 10 min followed by 40 cycles of 95 °C for 15 s and 60 °C for 1 min. The 2^-ΔΔCT^ method was used for relative quantification. Changes in gene expression were expressed as fold change.

### 2.8. Western Blotting

Liver tissue was homogenized in a buffer containing 10 mM Tris-HCl (pH 7.5), 150 mM NaCl, 2 mM EDTA, 1% Triton X-100, 10% glycerol, and a protease inhibitor cocktail (Roche, Barcelona, Spain) and kept on ice for 20 min. Homogenates containing 100 µg of protein were mixed in Laemmli loading buffer, boiled for 5 min, separated on a sodium dodecyl sulfate (SDS) 10% poly-acrylamide gel electrophoresis and transferred onto polyvinylidene fluoride (PVDF) membranes (Bio-Rad Laboratories Hercules, CA, USA).

After assessing transfer, membranes were incubated with blocking solution (5% non-fat milk and 1% Tween 20 in Tris-buffered saline, TBS) and incubated overnight at 4 °C with primary antibodies against the following proteins: Protein kinase B (Akt (pan) C67E7, 1:1000, Cell Signaling, Danvers, MA, USA); Phospho-Akt (Ser 473) (D9E) XP, 1:1000, Cell Signaling, Danvers, MA, USA); Nuclear factor kappa-light-chain-enhancer of activated B cells (NF-κB bs-0465R, 1:1000, Beijing Biosynthesis Biotechnology Company, P.R. China); Phospho-NF-κB p65 (s536), 1:1000, Cell Signaling, Danvers, MA, USA), and 70 kDa heat shock protein as internal control (Hsp-70 sc-7298, 1:500; Santa Cruz, CA, USA). Membranes were digitally imaged and quantified with scanning densitometry relied on ImageLab software version 6.0.1 (Bio-Rad Laboratories, Hercules, CA, USA).

### 2.9. Biochemical Determinations

Circulating myeloperoxidase (MPO); E-selectin; soluble intercellular adhesion molecule-1 (sICAM1); granulocyte-macrophage colony-stimulating factor (GM-CSF); regulated upon activation, normal T cell expressed and presumably secreted (RANTES); macrophage inflammatory protein-alpha (MIP-1 α); monocyte chemoattractant protein-1 (MCP-1); interleukin (IL)-18; interferon-gamma (INF-γ); IL-1α; IL-12; IL-4; IL-10, and IL-13 were measured in serum using MILLIplex^TM^ immunoassays (Merck-Millipore, MA, USA) and the Luminex 200 system.

### 2.10. Lipid Peroxidation

Malondialdehyde (MDA) was determined by the thiobarbiturate reaction [25]. For this purpose, 1 mL of trichloroacetic acid (20%) was added to 1 mL of homogenate in phosphate buffer 0.067 M at pH 7.4. After mixing and centrifuging, 1 mL of thiobarbiturate (0.67%) was added to the supernatant and boiled for 60 min. After cooling, optical density at 530 nm was assayed. 

### 2.11. Neutrophil Infiltration

Hepatic myeloperoxidase (MPO) activity was measured as a biomarker of neutrophil infiltration and activation. Two hundred µg of cryolyophilized liver was resuspended in 1.5 mL of 0.5% hexadecyltrimethylammonium bromide in 50 mM phosphate buffer, pH 6, and mechanically lysed using a vortex for 15 min. Homogenates were subjected to 3 cycles of freezing in liquid nitrogen/thawing at room temperature. Samples were then incubated during 2 h at 60 °C and centrifuged at 4000× *g* for 12 min. Supernatants were collected for MPO activity evaluation. Solutions containing 10 µL of supernatant, 10 µL of tetramethylbenzidine (work concentration 1.6 mM) in dimethyl sulfoxide, and 70 µL of H_2_O_2_ (work concentration 3.0 mM) in 80 mM phosphate buffer pH 4.5 were measured photometrically at 630 nm every 15 s for a total of 4 min [26].

### 2.12. Hepatic LPS

LPS was measured in liver homogenates using a competitive inhibition enzyme immunoassay (CEB526Ge, Cloud-CloneCorp., Houston, TX, USA) following the manufacturer’s protocol.

### 2.13. Statistical Analysis

One-way ANOVA followed by Dunnett’s multiple comparisons test was performed using GraphPad Prism version 8.0.1 for Windows (GraphPad Software, San Diego, CA, USA). *p* < 0.05 was considered statistically significant. All results were expressed as mean ± standard error of the mean (SEM). 

## 3. Results

In this work, we used Zucker-Lepr*^fa/fa^* rats as a model of insulin resistance syndrome (IRS) [24]. These rats manifest the liver component of IRS, NAFLD, among others [23]. At time 0, rats exhibited a lean phenotype with a weight of 179.9 ± 2.2 g, whereas after 30 days of intervention with the placebo the weight was 294.4 ± 5.7 g. Probiotic treatments did not modify rats’ body weight. Body weights were 282.1 ± 11.5 g in the *B. breve* group, 292.4 ± 10.8 g in the *L. paracasei* group, and 295.8 ± 13.0 g in the *L. rhamnosus* group.

### 3.1. Lactobacillus paracasei CNCM I-4034, Bifidobacterium breve CNCM I-4035 and Lactobacillus rhamnosus CNCM I-4036 Modulated Circulating Levels of Inflammation-Related Mediators in Zucker-Lepr^fa/fa^ Rats

Liver inflammation is regulated by chemokines, which modulate the migration and activities of hepatocytes, Kupffer cells, hepatic stellate cells, endothelial cells, and circulating immune cells [27]. Serum levels of GM-CSF, RANTES, MIP-1 α, and MCP-1 were determined (Figure 1). Both MIP-1 α and MCP-1 increased in the serum of Zucker-Lepr*^fa/fa^* rats that received *L. paracasei* CNCM I-4034 for 30 days in comparison with the placebo group (Figure 1C,D). Similar findings in these two chemokines and GM-CSF were induced by *L. rhamnosus* CNCM I-4036, although the differences with the placebo group were not statistically significant (Figure 1A,C,D).

Serum levels of two adhesion molecules, E-selectin and sICAM-1, were also measured (Figure 2). E-selectin is a specific endothelial adhesion receptor induced by pro-inflammatory stimuli. It mediates the adhesion of leukocytes to endothelial cells allowing their extravasation into inflammed tissues [28]. NAFLD is associated with elevated circulating levels of E-selectin and sICAM [29]. All groups exhibited similar values of E-selectin (Figure 2A). However, circulating sICAM-1 significantly increased in the rats treated with *L. paracasei* CNCM I-4034 (Figure 2B).

Circulating levels of pro-inflammatory (Figure 3) and anti-inflammatory (Figure 4) cytokines were also determined. Probiotics intake did not produce significant changes in serum IFN-γ, IL-1α and IL-12 compared with the placebo (Figure 3B–D). IL-18, however, increased in the serum of rats fed *L. paracasei* CNCM I-4034 (Figure 3A). Regarding anti-inflammatory cytokines, IL-4 increased in the *L. paracasei* group, although the difference with the placebo group did not reach statistical significance (*p* = 0.07, Figure 4A). IL-10 results are noteworthy (Figure 4C): although no statistical significance was reached, this IL tended to be higher with *L. paracasei* CNCM-I 4034 and *L. rhamnosus* CNCM-I 4036 treatments, the reason probably being that the number of rats (out of 7–8/group) where this cytokine was detected was much higher in the latter groups (control: 1 of 8 rats; placebo: 5 of 8; *B. breve*: 2 of 8; *L. paracasei*: 7 of 8; *L. rhamnosus*: 4 of 8).

### 3.2. Lactobacillus paracasei CNCM I-4034, Bifidobacterium breve CNCM I-4035 and Lactobacillus rhamnosus CNCM I-4036 Decreased Leukocyte Infiltration in the Liver of Zucker-Lepr^fa/fa^ Rats

Due to the central role of leukocytes in the development and propagation of inflammation, we evaluated the infiltration of leukocytes in the rats’ liver to determine the potential effect of the three probiotic strains (Figure 5). Liver tissue from the control group was normal, while Zucker-Lepr*^fa^*^/*fa*^ rats liver parenchyma clearly showed the accumulation of lipids in cytoplasmic vacuoles characteristic of macrovesicular hepatic steatosis (Figure 5, upper panels). All three probiotic strains diminished leukocyte infiltration in the liver of the Zucker-Lepr*^fa/fa^* rats, as shown in the graph representing quantitative data obtained from all stained liver sections (Figure 5, lower panel).

We also evaluated MPO activity in both liver tissue and serum, an enzyme that has been used as a marker of neutrophil infiltration and activation, and liver MDA as an indirect measure of lipid peroxidation (Figure 6). Liver MPO activity significantly increased after 30 days of feeding with the placebo (Figure 6A). Conversely, administration of *L. paracasei* CNCM I-4034 and *L. rhamnosus* CNCM I-4036 reduced MPO activity, reaching values of lean rats at time 0 (Figure 6A). Liver MPO tended to decrease with *B. breve* CNCM I-4035 (*p* = 0.112). MDA content decreased in the liver of rats fed *L. paracasei* CNCM I-4034 and tended to decrease with *L. rhamnosus* CNCM I-4036 (*p* = 0.0724) for 30 days (Figure 6C).

It is known that LPS may trigger the inflammatory process [30]. The loss of LPS-binding protein (LBP) is related to an attenuation of the liver-mediated inflammation [31]. We have described that treatment of Zucker-Lepr*^fa/fa^* rats with these three probiotic strains diminished the LBP concentration in serum [23]. We, therefore, determined the hepatic LPS content but found no differences among groups (Figure 6D).

### 3.3. Administration of Lactobacillus paracasei CNCM I-4034, Bifidobacterium breve CNCM I-4035 and Lactobacillus rhamnosus CNCM I-4036 Modulated Macrophage Polarization in Liver

Macrophages are recognized to exist as two distinct subtypes, M1 pro-inflammatory and M2 anti-inflammatory [6]. We investigated whether *L. paracasei* CNCM I-4034, *B. breve* CNCM I-4035 and *L. rhamnosus* CNCM I-4036 could affect the distribution of both subtypes of macrophages in the rats’ livers. We measured Cd86 and Nos2 mRNA levels as markers of M1 macrophages (Figure 7A,B), and Cd163 and Arg1 as M2 markers (Figure 7C,D) by qRT-PCR. Administration of the placebo resulted in an induction of Cd86 mRNA in the liver (Figure 7A), and feeding with all three probiotic strains for 30 days significantly down-regulated Cd86 expression (Figure 7A). We did not find differences in Nos2 (Figure 7B) and Cd163 (Figure 7C) among groups. Arg1 mRNA was up-regulated with *B. breve*, whereas administration of *L. paracasei* CNCM I-4034 and *L. rhamnosus* CNCM I-4036 down-regulated Arg1 in the liver (Figure 7D). 

Activation of Akt is induced by different types of stress and provides a cell survival signal [32]. In addition, increased production of butyrate by probiotics has been associated with an amelioration of NAFLD through, among others, the activation of Akt [33]. Moreover, whereas NF-kB is known to promote inflammation, probiotics have been reported to counteract its activation [34]. We evaluated the ability of the probiotic strains to activate Akt and NF-kB by determining phosphorylared-Akt/Akt ratio, NF-kB and phosphorylated-NF-kB protein levels by Western blotting (Figure 8). Whereas *B. breve* CNCM I-4035 and *L. rhamnosus* CNCM I-4036 had no effects on P-Akt/Akt, P-NF-kB or NF-kB, *L. paracasei* CNCM I-4034 significantly increased the ratio P-Akt/Akt and NF-kB protein levels.

## 4. Discussion

Probiotics are live microorganisms that, when consumed in adequate amounts, confer a health effect on the host. Beneficial effects of probiotics have been reported in allergy, intestinal-related diseases, chronic liver disease, urinary tract infections, and respiratory infections, among others. Lactobacilli and bifidobacteria are the genera most frequently used as probiotics, and they exert their benefits through a variety of mechanisms [23].

We have reported that the administration of *Lactobacillus paracasei* CNCM I-4034, *Bifidobacterium breve* CNCM I-4035 and *Lactobacillus rhamnosus* CNCM I-4036 to Zucker-Lepr*^fa/fa^* rats attenuates the accumulation of fat in the rats’ liver and exerts anti-inflammatory effects such as lower serum concentrations of tumor necrosis factor alpha (TNF-α), IL-6 and bacterial LPS [24]. Also, expression of three genes (*Adamdec1*, *Ednrb* and *Ptgs1*/*Cox1*) was up-regulated in the intestinal mucosa of the obese rats compared with the lean rats [23]. This effect was in part mediated by a decrease in both macrophage and dendritic cell populations. Probiotic treatment also increased secretory IgA content and diminished the LBP concentration [24].

Inflammation is a hallmark of liver disease. The immune response plays a key role in initiation, progression and resolution of hepatic inflammation [35]. A great body of evidence points that regulation of the phenotype of resident hepatic macrophages, Kupffer cells, is related to the progression of liver diseases such as alcoholic liver disease, NAFLD, non-alcoholic steatohepatitis (NASH), acute liver injury, alcoholic hepatitis, acute liver failure and hepatocellular carcinoma [35].

In this work we used the NAFLD model of Zucker-Lepr*^fa/fa^* rats to investigate the immune response associated with the administration of these three bacterial strains in hepatic macrophage modulation and their effects in liver damage. Overall, after 30 days of intervention and in comparison with a placebo, we observed an amelioration of hepatic leukocyte infiltration and damage markers that might be associated, at least in part, with their ability to modulate the immune response and macrophage distribution, although the effects were strain-dependent.

Secreted chemokines have a high affinity to glycosaminoglycans bound to extracellular matrix and endothelial surface. This property favors the local immobilization and retention of chemokines, creating a concentration gradient that allows a coordinated trafficking of leukocytes toward injured tissue [27]. In this regard, MCP-1 is associated with progression of simple steatosis to NASH, while RANTES is mainly involved in migration of T cells, monocytes, neutrophils, and dendritic cells through binding to its cognate transmembrane receptors [10], and is involved in several chronic immune-inflammatory diseases [36]. RANTES and MIP-1 α, among others, are up-regulated in fibrotic livers [37]. 

We found increased serum MIP-1 α and MCP-1 levels after *L. paracasei* CNCM I-4034 administration compared with the placebo group. It is worthy to note that, in spite of the ameliorated NAFLD-associated liver injury, we found increased pro-inflammatory mediators in the *L. paracasei* group, which seems to disagree with the improvement in liver damage and steatosis (the *B. breve* group was the one that exhibited the lower leukocyte hepatic infiltration). However, this is not the first time that high pro-inflammatory markers are described for other probiotic strains. For instance, similar findings have been reported for *L. paracasei* CNCM I-1518 [38]. Likewise, we have previously described elevated pro-inflammatory factors (TNF-α, IL-8 and RANTES) in dendritic cells cultured in the presence of *B. breve* CNCM I-4035 [39], which implies that the balance of both types of cytokines is essential to promote an adequate immune response. Thus, a massive recruitment of immune cells in the lungs due to administration of *L. paracasei* CNCM I-1518 strain and prior to influenza infection led to increased pro-inflammatory cytokine release [38]. This pre-activation state of the immune system seems to be responsible for the faster clearance of the infection, supporting the hypothesis that an early beneficial induction of the pro-inflammatory response before influenza infection, and a lower inflammatory response after this infection, is necessary to counteract an overactive immune response.

Administration of *L. paracasei* CNCM I-4034 induced NF-kB protein levels but no differences were found in its activated form. *L. paracasei* CNCM I-4034 also activated the Akt pathway, which is related to cell survival after different types of stress [32]. Furthermore, butyrate-producing probiotics have been associated with a reduction in the progression of fatty liver disease in rats through, among others, activation of Akt [33].

Specific adhesion glycoproteins are involved in the binding of leukocytes to endothelial cells [40]. NAFLD is associated with elevated circulating levels of E-selectin and sICAM [29,41]. We did not find differences in either E-selectin or hepatic LPS among the experimental groups, which suggests that changes in hepatic liver inflammation were not associated with changes in hepatic LPS. Circulating sICAM1 increased in the *L. paracasei* group. Although supposedly deleterious, the rise in this adhesion molecule together with the rise in the aforementioned chemokines may have been overcome by other protective mechanisms and for that reason the global effect in the liver was an improvement in this group. 

We also found a modulatory effect on the expression of hepatic macrophages after probiotic administration. Cd86 mRNA levels were down-regulated by all three probiotic strains. Decreased expression of M1 pro-inflammatory marker Cd86 was accompanied by the improvement in liver damage evaluated by MPO activity and leukocyte infiltration. These findings are consistent with the reductions in IL-6 and, TNF-α, and the improvement in steatosis that we have previously reported for these probiotic strains [24]. Also, *B. breve* CNCM I-4035 in particular up-regulated Arg1 mRNA expression, a marker of the M2 anti-inflammatory phenotype.

In conclusion, here we report that *B. breve* CNCM I-4035, *L. paracasei* CNCM I-4034 and *L. rhamnosus* CNCM I-4036 modulated liver macrophage gene expression in the Zucker-Lepr*^fa/fa^* rat NAFLD model, which was accompanied by an improvement in hepatic leukocyte infiltration. Effects on hepatic lipid peroxidation and damage was strain-dependent. In particular *L. paracasei* CNCM I-4034 administration resulted in improvements in hepatic damage and peroxidation. Although the link between macrophage shift and hepatoprotection by probiotics should be further investigated, its confirmation would support the idea that probiotics may be useful as a coadjuvant treatment for pathologies such as NAFLD.

## Figures and Tables

**Figure 1 nutrients-13-00202-f001:**
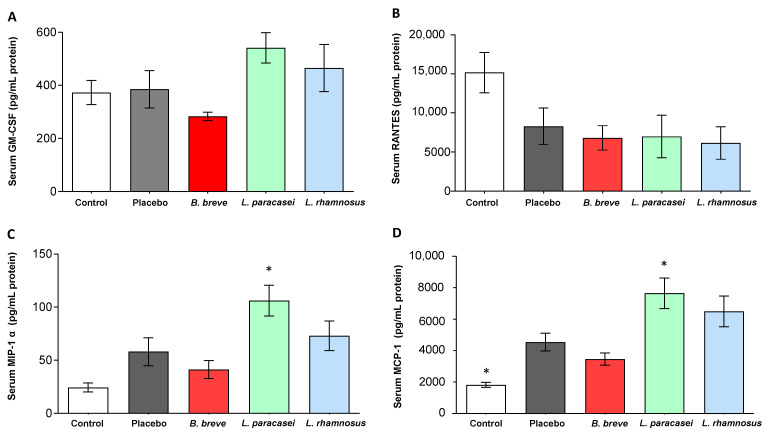
Serum concentration of (**A**) GM-CSF, (**B**) RANTES, (**C**) MIP-1 α, and (**D**) MCP-1 of rats fed either placebo, *B. breve* CNCM I-4035, *L. paracasei* CNCM I-4034 or *L. rhamnosus* CNCM I-4036 for 30 days. Values are the means ± SEM. *n* = 8 per group. * *p* < 0.05 vs. placebo. *B, Bifidobacterium*; GM-CSF, granulocyte-macrophage colony-stimulating factor; *L, Lactobacillus*; MCP-1, monocyte chemoattractant protein-1; MIP-1 α, macrophage inflammatory protein 1-alpha;RANTES, Regulated upon Activation, Normal T Cell Expressed and Presumably Secreted.

**Figure 2 nutrients-13-00202-f002:**
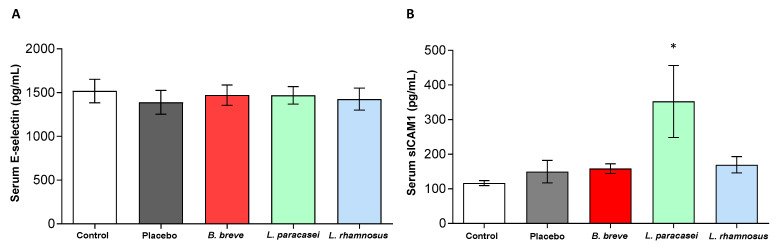
Serum concentration of (**A**) E-Selectin and (**B**) sICAM1 of rats fed either placebo, *B. breve* CNCM I-4035, *L. paracasei* CNCM I-4034 or *L. rhamnosus* CNCM I-4036 for 30 days. Values are the means ± SEM. *n* = 8 per group. * *p* < 0.05 vs. placebo. *B, Bifidobacterium*; *L, Lactobacillus*; sICAM1: soluble intercellular adhesion molecule-1.

**Figure 3 nutrients-13-00202-f003:**
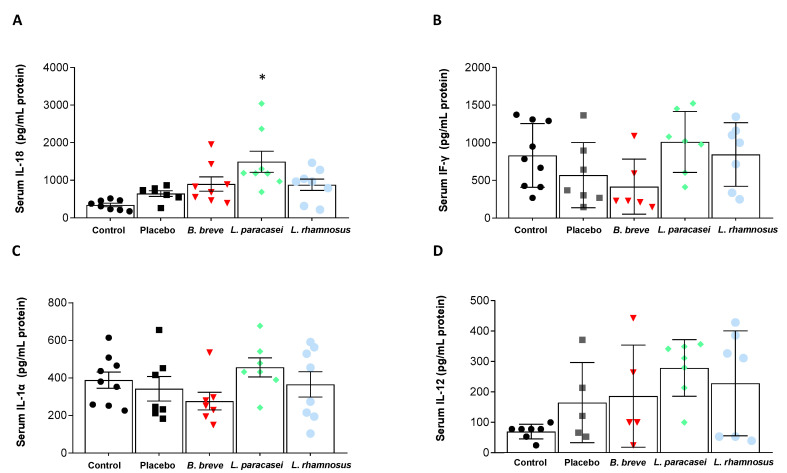
Serum concentration of pro-inflammatory cytokines. (**A**) IL-18, (**B**) INF-γ, (**C**) IL-1α, and (**D**) IL-12 of rats fed either placebo, *B. breve* CNCM I-4035, *L. paracasei* CNCM I-4034 or *L. rhamnosus* CNCM I-4036 for 30 days. Values are the means ± SEM. *n* = 8 per group. * *p* < 0.05 vs. placebo. *B, Bifidobacterium*; INF-γ, interferon-gamma; IL, interleukin; *L. Lactobacillus*.

**Figure 4 nutrients-13-00202-f004:**
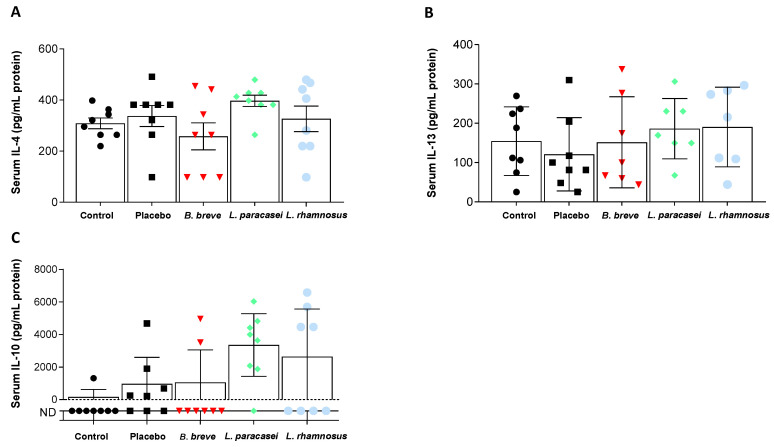
Serum concentration of anti-inflammatory cytokines. (**A**) IL-4, (**B**) IL-13, and (**C**) IL-10 of rats fed either placebo, *B. breve* CNCM I-4035, *L. paracasei* CNCM I-4034 or *L. rhamnosus* CNCM I-4036 for 30 days. Values are the means ± SEM. *n* = 8 per group. *B, Bifidobacterium*; IL, interleukin; *L. Lactobacillus*.

**Figure 5 nutrients-13-00202-f005:**
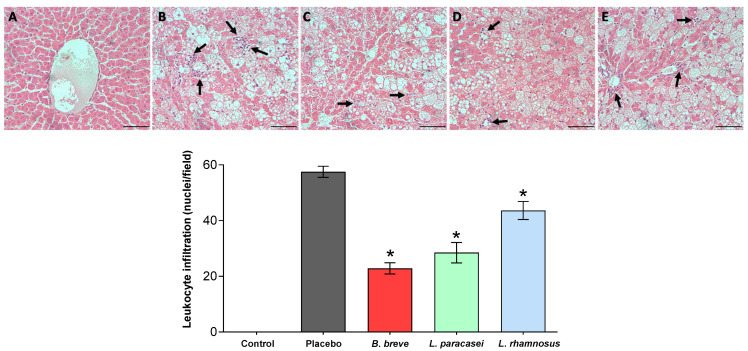
Inflammatory cell infiltration. Upper panels: Representative micrographs of 5-µm-thick liver sections stained with hematoxylin and eosin (H&E) of (**A**) control group, (**B**) placebo group, (**C**) *B. breve* group, (**D**) *L. paracasei* group, and (**E**) *L. rhamnosus* group. Bars = 20 µm. Lower panel: Quantification corresponding to the stained sections. Values are the means ± SEM, *n* = 3–4 per group. * *p* < 0.05 vs. placebo. *B, Bifidobacterium; L, Lactobacillus.*

**Figure 6 nutrients-13-00202-f006:**
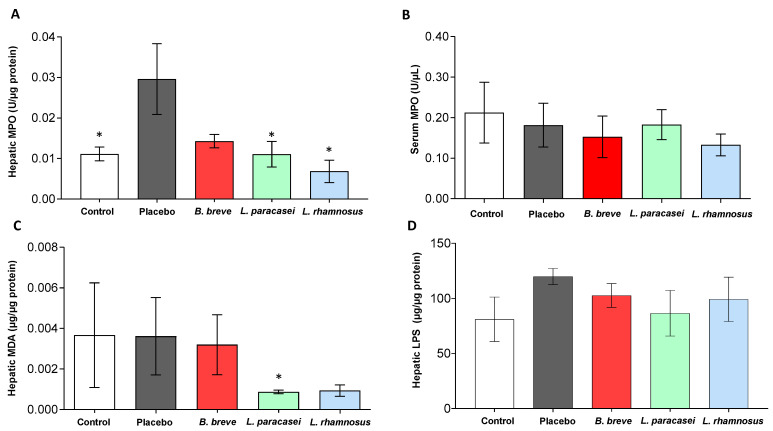
Hepatic levels of (**A**) MPO, (**C**) MDA and (**D**) LPS, and serum levels of MPO (**B**) of rats fed either with placebo, *B. breve* CNCM I-4035, *L. paracasei* CNCM I-4034 or *L. rhamnosus* CNCM I-4036 for 30 days. Values are the means SEM, *n* = 3–5 per group. * *p* < 0.05 vs. placebo. *B, Bifidobacterium; L, Lactobacillus*; LPS, lipopolysaccharide; MDA, malondialdehyde; MPO, myeloperoxidase.

**Figure 7 nutrients-13-00202-f007:**
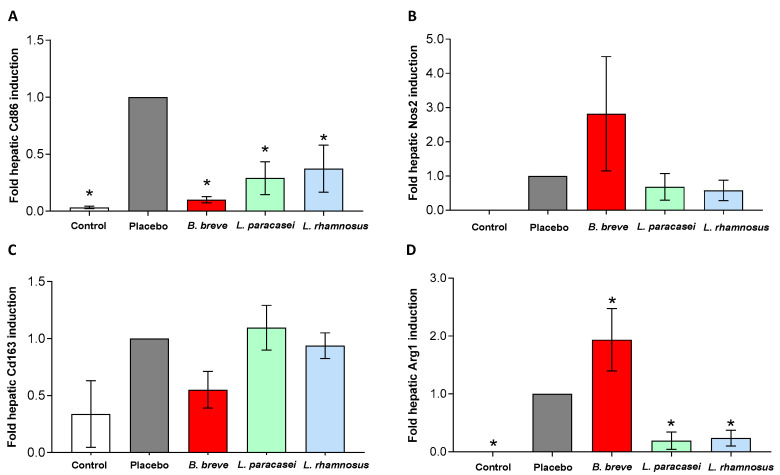
Expression of macrophage genes in liver. M1 macrophage markers: (**A**) Cd86 and (**B**) Nos2. M2 macrophage markers: (**C**) Cd163 and (**D**) Arg1 of rats fed either placebo or *B. breve* CNCM I-4035, *L. paracasei* CNCM I-4034 or *L. rhamnosus* CNCM I-4036 for 30 days. Relative mRNA levels were normalized to the trancript levels of the housekeeping gene Hprt1. Data were calculated as fold change compared with the placebo group. Values are the means ± SEM, *n* = 3–5 per group. * *p* < 0.05 vs. placebo. Arg1, Arginase-1; *B, Bifidobacterium*; Cd, cluster of differentiation; *L, Lactobacillus*; Nos2, Nitric oxide synthase-2.

**Figure 8 nutrients-13-00202-f008:**
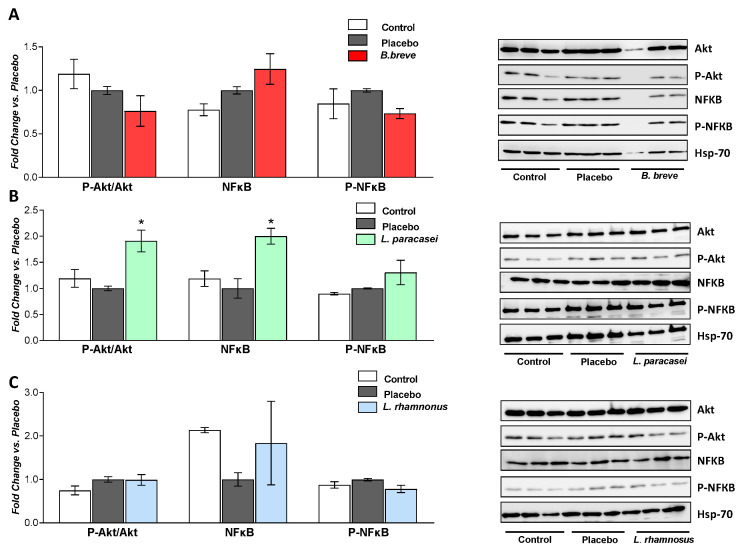
Western blot analysis of P-Akt/Akt ratio, NF-kB and P-NF-kB proteins of rats fed either placebo, (**A**) *B. breve*, (**B**) *L. paracasei* or (**C**) *L. rhamnosus* for 30 days. Hsp-70 was used as a loading control. Graphs on the left included 3 rats per group, and the right panels show representative blots. Values are the means ± SEM. * *p* < 0.05 vs. placebo. Akt, protein kinase B; *B, Bifidobacterium*; Hsp70, 70 KD heat shock protein; *L, Lactobacillus*. NF-κB, nuclear factor kappa-light-chain-enhancer of activated B cells; P-Akt, phosphorylated protein kinase B; P-NF-kB, phosphorylated nuclear factor kappa-light-chain-enhancer of activated B cells.

## Data Availability

The data presented in this study are available on request from the corresponding author.

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
