# Peer review of "Bifidobacterium breve CNCM I-4035, Lactobacillus paracasei CNCM I-4034 and Lactobacillus rhamnosus CNCM I-4036 Modulate Macrophage Gene Expression and Ameliorate Damage Markers in the Liver of Zucker-Leprfa/fa Rats"

_nutrients, 2021, doi:10.3390/nu13010202_

Round 1

Reviewer 1 Report

The authors present a manuscript studying the impact of treatment with several probiotic strains on hepatic macrophage polarization. While there is some interesting data presented here, the evidence does not fully support the claim that these probiotic strains are providing hepatoprotection by driving transition of macrophages from a proinflammatory to anti-inflammatory state. At this point the data are too preliminary to support a clear mechanism for the hepatoprotection previously observed with these three probiotic strains.  I would recommend the following:

  1. The authors note in figure 1 that mip-1 and mcp-1 are similarly increased after L. rhamnosus but not statistically significant. I would recommend expanding on this.  What is the p-value?  Does it fall within the range of being a trend (p=0.05-0.1)?
  2. The authors note that sICAM is significantly increased but there is no notation of this in Figure 2B. Is this difference statistically significant?
  3. In the evaluation of leukocyte infiltration, nuclei on H&E staining were counted as the only measure. Would recommend a more robust measure using specific targets via immunohistochemistry and quantification of positive cells (CD45, Mac-2, etc.). 
  4. The authors note that hepatic LPS is unchanged but in prior studies have shown that treatment with these strains of probiotics that serum LPS is significantly decreased. Is this due to a difference in the source being measured (serum vs. hepatic)?  Or is there a different explanation for the discrepancy between this data and that which was reported previously?
  5. In terms of macrophage polarization, the authors show a clear decrease in cd86 expression in the liver after treatment with all three strains. However, there does not appear to be in a consistent increase in markers of M2 macrophages except for Arg1 after B. breve.  The claim is being made that the macrophages are switching from a proinflammatory state to an anti-inflammatory state after treatment.  While the loss of CD86 is clear, the evidence is not strong to support an increase in M2 polarization. Would recommend evaluation of additional makers or provide further evidence of M2 polarization.
  6. Throughout manuscript, data for each probiotic strain are shown on the same graph but are separated out in figure 8. This is likely due to the limitation of being able to run a certain number of samples on a gel, but it appears that the control and placebo vary quite a bit from gel to gel.  This is particularly true for NFkB.  Would consider normalizing the data to the control or placebo so that each is presented as a fold change compared to control or placebo.
  7. Consider isolating macrophages so they can be studied separately from the rest of the hepatic tissue as this may reduce variability in some of the measurements being completed related to macrophage polarization.
  8. The authors previously showed that a mixture of the three probiotic strains had a more robust effect on histologic evidence of steatosis. Consider evaluating the impact of the combination of all three on macrophage polarization.  This may provide a more robust and consistent difference.
  9. Consider providing actual p-values on graphs or in text to assist with confusion between data that is statistically significant and those that appear to be different but not significantly so.

Author Response

Reviewer 1

The authors present a manuscript studying the impact of treatment with several probiotic strains on hepatic macrophage polarization. While there is some interesting data presented here, the evidence does not fully support the claim that these probiotic strains are providing hepatoprotection by driving transition of macrophages from a proinflammatory to anti-inflammatory state. At this point the data are too preliminary to support a clear mechanism for the hepatoprotection previously observed with these three probiotic strains.

We thank the reviewer for her/his constructive comments and review. Please keep in mind that we have been given 8 days to resubmit the manuscript, so this makes it undoable to carry out the additional measurements suggested by the reviewer. We then have toned down our statements regarding the link between hepatoprotection and macrophage shifting. We have tried to address you concerns below.

I would recommend the following:

The authors note in figure 1 that mip-1 and mcp-1 are similarly increased after L. rhamnosus but not statistically significant. I would recommend expanding on this. What is the p-value? Does it fall within the range of being a trend (p=0.05-0.1)?

We try to be categorical, so throughout the entire manuscript whenever we state that a particular change was statistically significant it means that the P value was ≤ 0.05. Likewise, when a change tended to be higher or lower it means that the P value was close to 0.05. In figure 1, the following differences were significant:

MCP-1: P=0,049 placebo vs control; P=0.021 placebo vs L. paracasei.

MIP1: P=0’023 placebo vs L. paracasei.

The phrase “Similar findings in these two chemokines and GM-CSF were induced by L. rhamnosus CNCM I-4036, although the differences with the placebo group were not statistically significant (Fig. 1A, 1C and 1D)” means that treatment with this strain resulted in differences that, compared with the placebo, did not reach significance. In fact, P values were higher than 0.2 for these 3 parameters and therefore there was not even a trend. We would rather not include these P values in the sentence. If the reviewer thinks it is best to delete this phrase we may do so.

The authors note that sICAM is significantly increased but there is no notation of this in Figure 2B. Is this difference statistically significant?

In this particular case, an asterisk “disappeared” when trying to insert the original figure into the manuscript. The figure has been corrected. This difference was statistically significant.

In the evaluation of leukocyte infiltration, nuclei on H&E staining were counted as the only measure. Would recommend a more robust measure using specific targets via immunohistochemistry and quantification of positive cells (CD45, Mac-2, etc.).

We agree with the reviewer that the suggested quantification would strengthen our statements but keep in mind that we have been given 8 days to revise the manuscript. It is impossible for us to meet the deadline.

The authors note that hepatic LPS is unchanged but in prior studies have shown that treatment with these strains of probiotics that serum LPS is significantly decreased. Is this due to a difference in the source being measured (serum vs. hepatic)?  Or is there a different explanation for the discrepancy between this data and that which was reported previously?

The reviewer is correct: LPS values already published were measured in serum; LPS data presented in this manuscript were measured in liver tissue.

In terms of macrophage polarization, the authors show a clear decrease in cd86 expression in the liver after treatment with all three strains. However, there does not appear to be in a consistent increase in markers of M2 macrophages except for Arg1 after B. breve.  The claim is being made that the macrophages are switching from a proinflammatory state to an anti-inflammatory state after treatment.  While the loss of CD86 is clear, the evidence is not strong to support an increase in M2 polarization. Would recommend evaluation of additional makers or provide further evidence of M2 polarization.

Again we agree that additional markers of M2 macrophages could be studied, but not in a matter of 8 days. We have toned down our statements about the link between hepatic macrophage polarization and the improvement in liver damage. Please see lines 41-43, 342-345(discussion) and 390-397(conclusion).

Throughout manuscript, data for each probiotic strain are shown on the same graph but are separated out in figure 8. This is likely due to the limitation of being able to run a certain number of samples on a gel, but it appears that the control and placebo vary quite a bit from gel to gel. This is particularly true for NFkB. Would consider normalizing the data to the control or placebo so that each is presented as a fold change compared to control or placebo.

The reviewer is correct about the reason for the different layout of this figure. As suggested, we have normalized the results of this figure to the placebo.

Consider isolating macrophages so they can be studied separately from the rest of the hepatic tissue as this may reduce variability in some of the measurements being completed related to macrophage polarization.

We do not have experience in macrophage isolation so this would be very difficult and take us definitely longer than 8 days.

The authors previously showed that a mixture of the three probiotic strains had a more robust effect on histologic evidence of steatosis. Consider evaluating the impact of the combination of all three on macrophage polarization.  This may provide a more robust and consistent difference.

Actually, the mixture used in previous papers was a combination of L. paracasei and B. breve, not a combination of the 3 strains. We decided not to use the mixture for this work.

Consider providing actual p-values on graphs or in text to assist with confusion between data that is statistically significant and those that appear to be different but not significantly so.

We have included the P value whenever there is a statistical trend: line 243 and lines 274-278.

Reviewer 2 Report

This is a very interesting study examining the effects of 3 different probiotics on macrophage polarization, inflammation and liver damage in obese Zucker fatty rats. I do have several suggestions for improvement of the manuscript.

1) Page 5 lines 206 and 207, the authors provide body weight information only about the placebo group. The authors need to provide initial and final body weight information for all experimental groups.

2) The authors need to provide data on the effect of the probiotics on hepatic steatosis either through Oil Red O measurement of hepatic lipids or hepatic triglyceride levels.

3) It is not clear how the different probiotic strains were administered. Were they administered by oral gavage or by capsule? 

Author Response

Reviewer 2

This is a very interesting study examining the effects of 3 different probiotics on macrophage polarization, inflammation and liver damage in obese Zucker fatty rats. I do have several suggestions for improvement of the manuscript.

We thank the reviewer for her/his kind comments. We have addressed your concerns below.

1) Page 5 lines 206 and 207, the authors provide body weight information only about the placebo group. The authors need to provide initial and final body weight information for all experimental groups.

Average body weight ± SEM for all study groups appear on lines 211-213.

2) The authors need to provide data on the effect of the probiotics on hepatic steatosis either through Oil Red O measurement of hepatic lipids or hepatic triglyceride levels.

We did both an Oil Red O staining of liver sections and trygliceride measurements in liver tissue. These results were published in one of our previous papers. Please see reference #24 (Doi: 10.1371/journal.pone.0098401).

3) It is not clear how the different probiotic strains were administered. Were they administered by oral gavage or by capsule?

Both probiotics and placebo were administered by oral gavage. We have clarified this issue (lines 127-134).

Reviewer 3 Report

The topic of the manuscript is of first importance. In my opinion, the study has been properly designed and described. The results of the study are very interesting and promising. I recommend to consider the manuscript to be published after minor revision done. Details are below:

1.In figure legends, all abbreviations should be explained independently of the main text. The study groups should also be described in figure legends.

2. The first paragraph of 3.2 subchapter should be rewritten - it is unclear.

3. The references - a citation style should be unified in the list (journal abbreviations, etc). Positions 17 and 40 should be corrected.

4. There are a few minor errors in the English language that need to be corrected in the manuscript.

Author Response

Reviewer 3

The topic of the manuscript is of first importance. In my opinion, the study has been properly designed and described. The results of the study are very interesting and promising. I recommend to consider the manuscript to be published after minor revision done. Details are below.

We thank the reviewer for her/his kind comments. We have addressed your concerns below.

  1. In figure legends, all abbreviations should be explained independently of the main text. The study groups should also be described in figure legends.

Abbreviations and groups have been added to figure legends, as requested.

  1. The first paragraph of 3.2 subchapter should be rewritten - it is unclear.

The first paragraph in section 3.2 (lines 258-264) has been clarified, as requested.

  1. The references - a citation style should be unified in the list (journal abbreviations, etc). Positions 17 and 40 should be corrected.

Citation style has been revised. Errors in references 17 and 40 have been corrected.

  1. There are a few minor errors in the English language that need to be corrected in the manuscript.

We have revised the manuscript and corrected a few misspellings.

Round 2

Reviewer 1 Report

Concerns that could be addressed in the limited time for resubmission have been done so appropriately.